# Higher-generation type III-B rotaxane dendrimers with controlling particle size in three-dimensional molecular switching

Chak-Shing Kwan[1], Rundong Zhao[2], Michel A. Van Hove [2], Zongwei Cai[1] & Ken Cham-Fai Leung [1,3]

Type III-B rotaxane dendrimers (T3B-RDs) are hyperbranched macromolecules with mechanical bonds on every branching unit. Here we demonstrate the design, synthesis, and characterization of first to third (G1–G3), and up to the fourth (G4) generation (MW > 22,000 Da) of pure organic T3B-RDs and dendrons through the copper-catalyzed alkyne–azide cycloaddition (CuAAC) reaction. By utilizing multiple molecular shuttling of the mechanical bonds within the sphere-like macromolecule, a collective three-dimensional contract-extend molecular motion is demonstrated by diffusion ordered spectroscopy (DOSY) and atomic force microscopy (AFM). The discrete T3B-RDs are further observed and characterized by AFM, dynamic light scattering (DLS), and mass spectrometry (MS). The binding of chlorambucil and pH-triggered switching of the T3B-RDs are also characterized by [1]H-NMR spectroscopy.

[1] Department of Chemistry and Partner State Key Laboratory of Environmental and Biological Analysis, Hong Kong Baptist University, Kowloon Tong, Kowloon, Hong Kong. [2] Department of Physics and Institute of Computational and Theoretical Studies, Hong Kong Baptist University, Kowloon Tong, Kowloon, Hong Kong. [3] Institute of Molecular Functional Materials, University Grants Committee, Hong Kong. Correspondence and requests for materials should be addressed to K.C.-F.L. (email: cfleung@hkbu.edu.hk)

The elegance of the mechanical bond between molecules is the possibility to build successive generations of increasingly sophisticated complex topological architectures[1–4] and molecular machines[5–10]. Rotaxanes[11–14], are mechanically interlocked molecules, that are prototypical components of molecular machines. For example, they can perform a linear one-dimensional (1D) molecular shuttling of their macrocycle along the rod in the discrete molecules. In contrast, a dendrimer[15–17] is a hyperbranched macromolecule with a well-defined mono-dispersed structure. The marriage between rotaxane and dendrimer creates a class of mechanically interlocked architectures known as rotaxane dendrimers (RDs)[18–20]. Among the three main types (I–III) of RDs, type III RDs are defined as dendritic polyrotaxanes and the mechanical bonds grow like dendrimers. The intrinsic complexity of type III-B RDs (T3B-RDs) and steric hindrance with hyperbranched mechanical bonds render the formation of higher generation (G > 2) T3B-RDs rather difficult, thus poses as a greater synthetic challenge than the type III-A RDs[21,22]. The earliest example of first generation (G1) T3B-RD was reported by Vögtle colleagues[23]. We reported the first facile synthesis of prototypical G2 type III-B rotaxane dendrons in 2013[24]. The preparation of higher generation T3B-RD has not been deciphered since then.

Herein, we report our comprehensive study toward the design, synthesis, characterization, and properties exploration of G1–G3 T3B-RDs and a G4 dendron. These molecules are pure organic compounds. In particular, owing to the sphere-like dendritic nature of T3B-RD, they could translate each [2]rotaxane with individual 1D molecular shuttling into a three-dimensional (3D) molecular size control within its multiple switchable rotaxane motif. This macromolecule could also induce an overall extension–contraction[2,8,25] or breathing molecular motion[26–29] via the polyrotaxane molecular shuttling of T3B-RD simultaneously. The size, polarity, and the accessibility of the void spaces in the T3B-RD can be controlled.

## Results

**Design and synthesis of type III-B RDs**. The design and synthesis of first to fourth generation (G1–G4) T3B rotaxane dendrons followed a convergent pathway, which modifies the functional groups (N-hydroxysuccinimide (NHS), acetylene, or azide) (Fig. 1 and Supplementary Figs. 1–25). Fréchet-type arylether dendrons were employed as the dendritic stoppering surface groups rendering the T3B-RDs readily soluble in most organic solvents (Fig. 2). Thread H·PF$_6$ which consists of a dibenzylammonium (DBA) bearing two acetylene groups, was first self-assembled with functionalized dibenzo-24-crown-8-OSu (DB24C8-OSu) to form a pseudorotaxane (Thread H·PF$_6$ ⊂ DB24C8-OSu), followed by a stoppering process with Fréchet-type dendrons through copper catalyzed azide–alkyne cycloaddition (CuAAC). The resulting first generation (G1) [2]rotaxane dendron-NHS was synthesized with a 82% yield. The activated NHS-ester on the macrocycle was then reacted with either the linker acetylene S6 or azide S14 to give the G1 [2]rotaxane azide dendron and G1 [2]rotaxane acetylene dendron, respectively. These two dendrons (G1 azide and G1-acetylene) were reacted through a CuAAC reaction to give the G1 [3]rotaxane dendrimer with a 83% yield. The synthetic methodology and the development of a robust route allows us to continue the synthesis of higher generation T3B-RDs by 1:1 molar equivalent of both starting dendron materials, and suitable linker core length to avoid excessive steric hindrance from the dendrons themselves. In the growing of the second generation, by repeating the same CuAAC reaction, G2 [7]rotaxane dendrimer was synthesized in 84% yield. In the synthesis of the third generation G3 [8]rotaxane

dendrons, the yield decreased to ~41%. G3 [15]rotaxane dendrimer was obtained via the same CuAAC reaction with a moderate yield (71%). To boost the limit of synthesizing T3B-RDs, a fourth generation (G4) dendron is presented as a prototypical example. This molecule involved even greater steric hindrance from G3 dendrons that could render the reaction unfavorable and also difficult to isolate the targeted product. With the promising CuAAC reaction, G4 [16]rotaxane dendron was finally successfully synthesized with an acceptable yield of 41%.

**Characterization of type III-B RDs**. All G1–G3 T3B-RDs and the G4 T3B [16]rotaxane dendron were characterized by [1]H-NMR and [13]C-NMR spectroscopy (Supplementary Figs. 26–44). By taking G1 [2]rotaxane dendron-NHS as an example (Supplementary Fig. 29), DBA $H_l$ on the thread shifted downfield ($\Delta\delta H_l = 0.51$ p.p.m.) and splited to a triplet because of the formation of hydrogen bonding with the oxygen atom on DB24C8. The aromatic protons $H_k$ and $H_j$ experienced an upfield shift with $\Delta\delta$ of −0.31 and −0.10 p.p.m. due to the shielding effect of DB24C8. All aromatic protons of the macrocycle ($H_q$, $H_r$, $H_p$, $H_m$, and $H_n$) shifted upfield, as a result of stabilizing effect by forming the hydrogen bonding with the thread. The proton and carbon signals of both G1 dendron, and G1 [3]rotaxane dendrimer were well identified with sharp signals. The disappearance of succinimide protons $H_m$ ($\delta = 2.77$ p.p.m.), and two new sets of linker signals ($H_m$, $H_p$) in the two dendrons were found, suggesting a complete exchange of the activated ester on the crown ether (Supplementary Fig. 26). The new triazole proton $H_w$ ($\delta = 7.70$ p.p.m.), $H_s$ and $H_{aa}$ in G1 [3]rotaxane dendrimer's spectrum indicated that the core was linked with new triazole formed by the two G1 azide and acetylene dendrons. Moreover, in G2, similar NMR proton signals were observed, except for the G2 [4]rotaxane dendrons azide (Supplementary Fig. 27). $H_{jj}$ next to the amide merged with the $H_t$ proton signals. In G2 [7]rotaxane dendrimer, the proton next to the amide merged to one peak. The proton signals in G3 [8]rotaxane dendrons were capable to be characterized by [1]H NMR. The integrations of new succinimide protons, amide protons, and acetylene proton were consistent with the calibrated value even with large differences in molecular weight (Supplementary Fig. 28). The [1]H-NMR signals of G3 [15] RDs became broader, but still with the same pattern. Thereby, it is consistent with the G1 and G2 RDs, with reasonable integrations. In view of stacked [1]H NMR of G1–G3 T3B-RDs (Fig. 3), peak broadening was increased sequentially because of the large number of repeating units and high-molecular weights. For G4 [16]rotaxane dendron, interestingly, the variation of peak area of each NMR signal became very large. Fortunately, the [1]H-NMR signal ($\delta = 2.72$ p.p.m., succinimide) of the new functionality NHS in G4 [16]rotaxane dendrons-NHS was able to identify. This result indicates that the compound was terminated with the correct functionalized DB24C8, and the new rotaxane moiety was successfully facilitated.

To further confirm the purity of the products, two-dimensional (2D) [1]H diffusion ordered spectroscopy (DOSY) was performed (Fig. 4). One set of signal was clearly observed from DOSY spectra for each G1–G3 T3B-RDs (Supplementary Figs. 33–35), indicating that only one component existed with high purity. Besides 1D NMR, 2D nuclear overhauser effect spectroscopy (NOESY) was used to characterize the position of the macrocycles within the T3B-RDs. Clear cross peaks between the crown ether aliphatic protons and the aromatic protons of DBA were observed in the NOESY spectra of G1–G3 (Supplementary Figs. 39–41), it showed that DB24C8 rings were located at the DBA sites, respectively.

HR-MALDI-TOF and HR-ESI mass spectrometries were the used to confirm all dendrons and dendrimers with their mass-to-charge ratios ($m/z$). HR-MALDI-TOF was performed on three

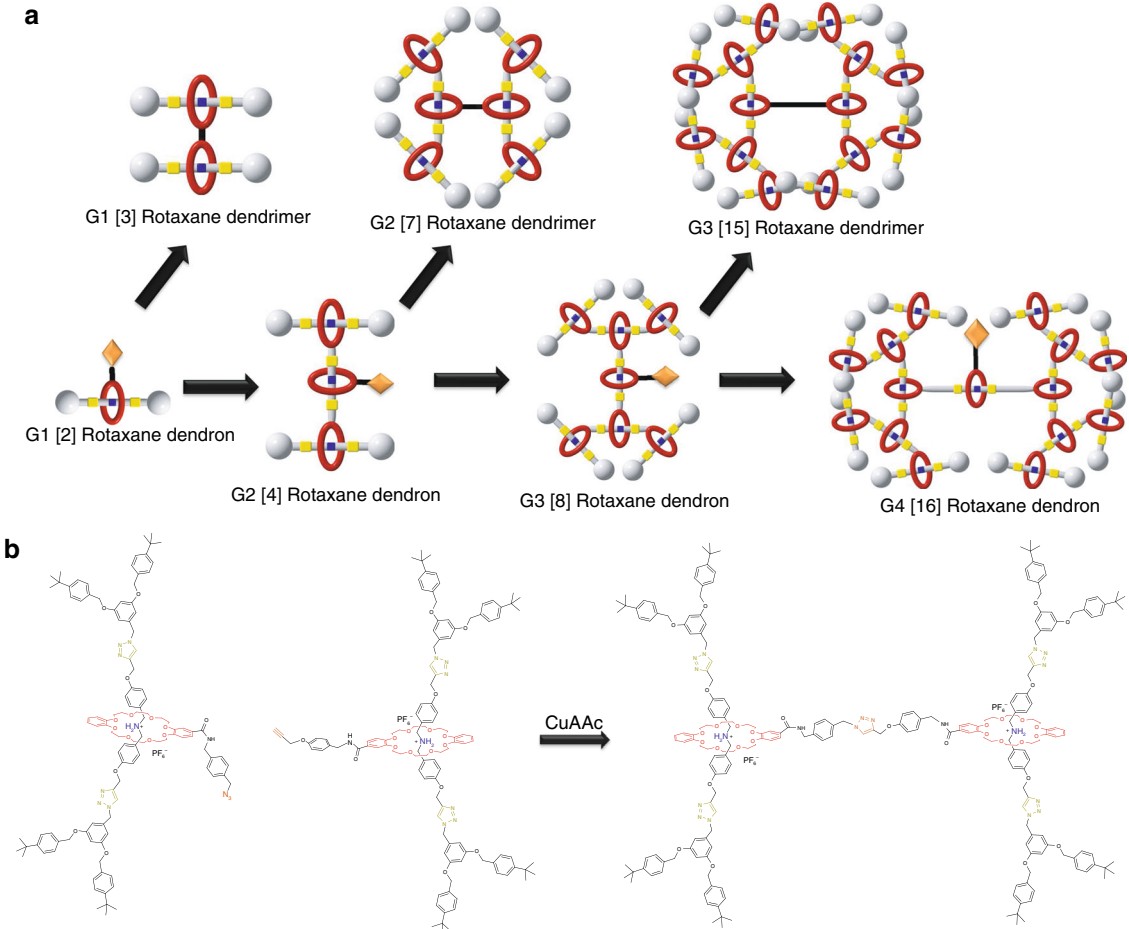

**Fig. 1** Sketches of type III-B rotaxane dendrimers. **a** Schematic diagram of the growing type III-B rotaxane dendrimers (T3B-RDs) via a convergent approach. **b** Synthesis of G1 [3]rotaxane dendrimer through copper catalyzed azide–alkyne cycloaddition (CuAAC) reaction. In each generation, a functionalized rotaxane dendrons terminated with N-hydroxysuccinimide (NHS) esters were synthesized, and exchange to azides/alkynes moieties. One new mechanical bond was formed in every dendrons synthesis. The two azides/alkynes dendrons join together via CuAAC to give RDs

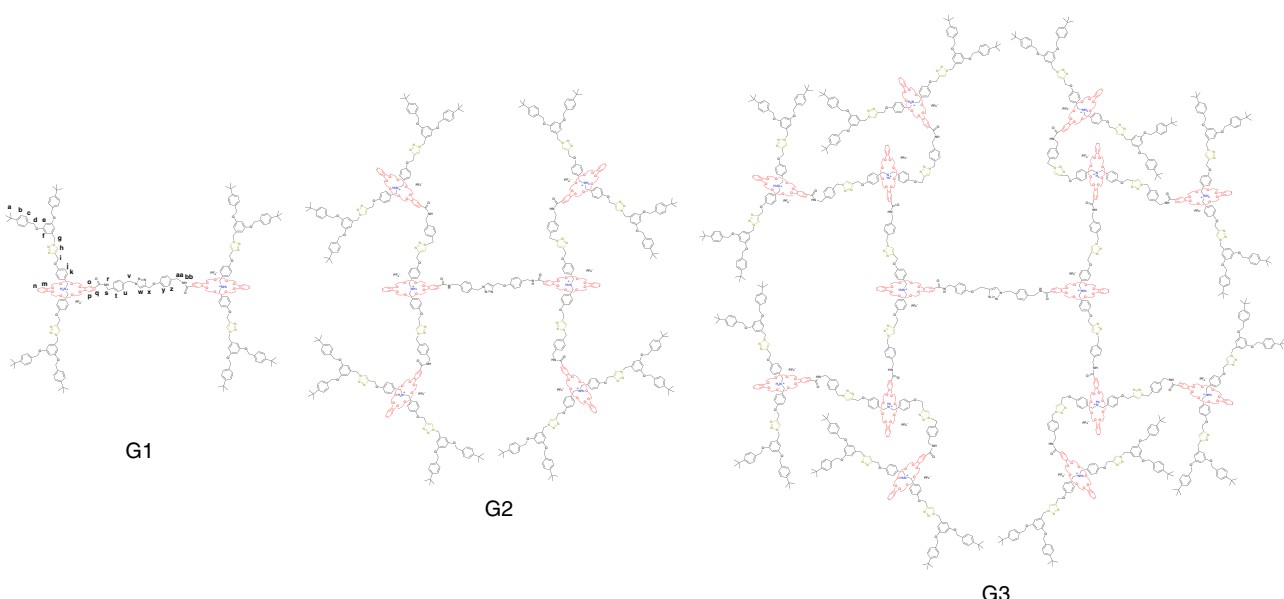

**Fig. 2** Molecular structures of type III-B rotaxane dendrimers. All generations of T3B-RDs composed with the same backbone. The mechanical bonds constituted the branching points and extended from the thread. The number of mechanical bonds increased from 2 in G1 [3]rotaxane dendirmer to 6 in G2 [7]rotaxane dendrimer, and 14 in G3 [15]rotaxane dendrimer

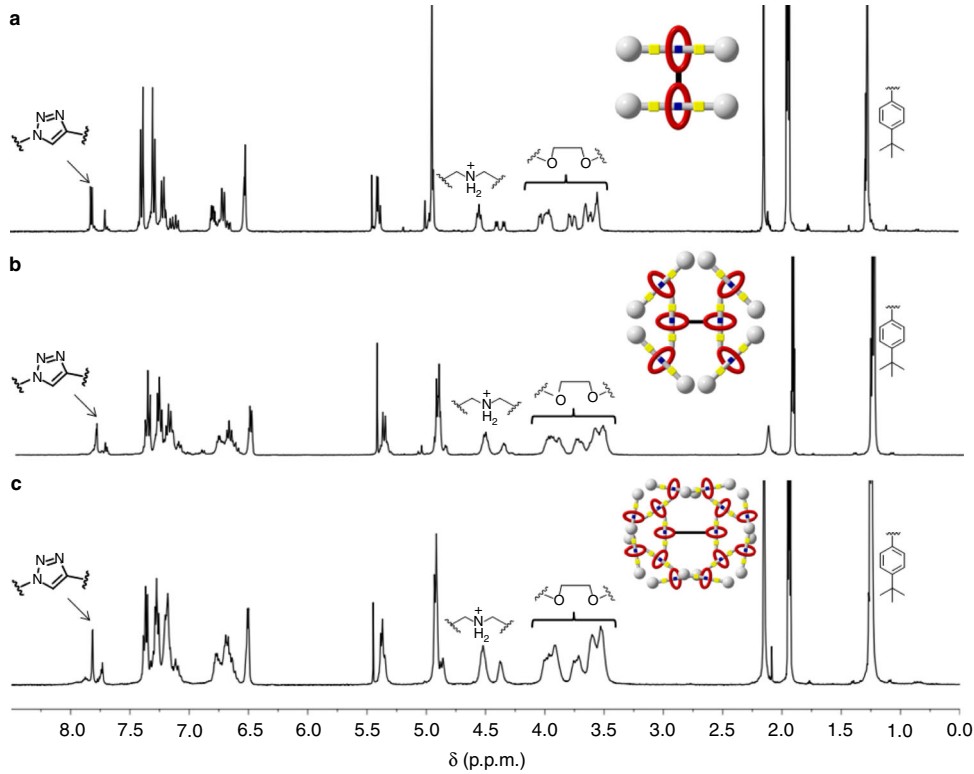

**Fig. 3** Stacked $^1$H-NMR spectra. All three generations of T3B-RDs showed the similar $^1$H-NMR peaks. **a–c** The representative peaks were labeled. The peak broadening was observed from the increase of generation. CD$_3$CN was the solvent used in these analyses

singly charged G1 dendrons. Starting from G1 dendrimers or higher generations, all compounds were multiple charged, thereby HR-ESI was chosen for the mass analysis. By way of an example for G1 [4]RDs, [M−2PF$_6$]$^{2+}$ ion was found ($m/z = 1857.4723$) in agreement with the theoretical value ($m/z = 1857.4788$). In all G2 dendron spectra, [M−3PF$_6$]$^{3+}$ ions were found confirming all targeted structures. Small rational deviation of $m/z$ [M−6PF$_6$]$^{6+}$ from theoretical value ($m/z =$ calcd 1552.9567) was found in G2 [7]RDs ($m/z = 1552.7983$) with an increased molecular weight and charge number. Moreover, G3 dendrons have even higher molecular weights and charges, and that similar rational deviation of $m/z$ was found. [M−13PF$_6$]$^{13+}$ ($m/z = 1588.96254$) signal was observed for G3 [15]rotaxane dendrimer. For G4 [16]rotaxane dendron-NHS, a clear [M−13PF$_6$]$^{13+}$ signal ($m/z = 1668.9801$) was found (Fig. 5), comparing to the calculated value ($m/z = 1669.9046$). With such a high-charge number and molecular weight, it indicates the largest G4 [16]rotaxane dendron was successfully synthesized and isolated after purification by column chromatography. All the G1–G3 T3B-RDs and G4 dendron were clearly identified in mass spectra and consistent with the theoretical value, confirming all the successful syntheses.

**Properties of type III-B RDs.** With the pH-responsive rotaxane components on the T3B-RDs, they were deprontonated and isolated to give the neutral G1–G3 RDs. All ammonium ions within the T3B-RDs were deprotonated by a phosphazene-based BEMP resin[30,31] to give the neutral G1–G3 T3B-RDs eventually, after a simple filtration from the reaction mixture. The physical properties of the deprotonated neutral G1–G3 T3B-RDs are quite different. Originally, all G1–G3 T3B-RDs were soluble readily in CH$_3$CN. However, all neutral G1–G3 T3B-RDs were insoluble in CH$_3$CN.

In CD$_2$Cl$_2$, the $^1$H-NMR spectra of all neutral G1–G3 T3B-RDs showed significant signal shifts in comparison to the original G1–G3 T3B-RDs. In the case of neutral G1 T3B-RD (Supplementary Fig. 56), the proton adjacent to DBA $H_l$ shifted upfield

significantly ($\Delta\delta H_l = −0.83$ p.p.m.), indicating that $H_l$ was not bonded with DB24C8's oxygen atom anymore. The aromatic protons of DBA shifted downfield, resulting of a deshielding effect after the switching of macrocycle. Those peak shifts directly confirmed that the ring was not located at the sec-amine site after deprotonation. Conversely, the triazole $H_h$ experienced a significant downfield shift ($\Delta\delta H_h = 0.48$ p.p.m.) and boarding owing to interaction of $H_h$ with DB24C8 through hydrogen bonding and the shuttling of the marcocycle between two triazoles. Meanwhile, $H_g$ signal became smaller, $H_i$ shifted upfield, and the aromatic protons on the dendrons shifted slightly upfield as the shielding effect of the macrocycle, confirming that the DB24C8 preferentially resided at the triazole rings after deprotonation. To further prove our hypothesis, 2D NOESY was performed (Fig. 6). Clear cross peaks of $H_h$ with the glycol protons of DB24C8 were found because of the correlation through space (Supplementary Fig. 42), while no cross peak was observed of the glycol protons with the middle triazole $H_w$. This NMR result showed that the ring was encircled on the triazole without the need of methylation. We also conducted density-functional-based tight binding (DFTB+) study, and the result showed that the DB24C8 tend to stay at triazole after the deprotonation of DBA (Fig. 7 and Supplementary Fig. 45–48). Deprotonated neutral G2 and G3 T3B-RDs showed similar peak shifts as of G1. The only difference was that the sec-amine peak was identified on G2 and G3 at $\delta = 4.68$ p.p.m. Same cross signals of $H_{triazole}$ were also observed in both G2 and G3 NOESY NMR (Supplementary Figs. 43 and 44). All the neutral T3B-RDs only show one set of signals, indicating that the shuttling of the macrocycle between the two triazoles was faster than the NMR timescale.

Besides 1D and 2D NMR, $^{31}$P NMR and MALDI-TOF were used to confirm the complete removal of PF$_6^−$. Before the treatment with BEMP resin, the signal ($\delta = −144$ p.p.m.) on $^{31}$P NMR corresponding to HPF$_6$ for all G1–G3 T3B-RDs. Once they were deprotonated, no signal can be found on $^{31}$P NMR,

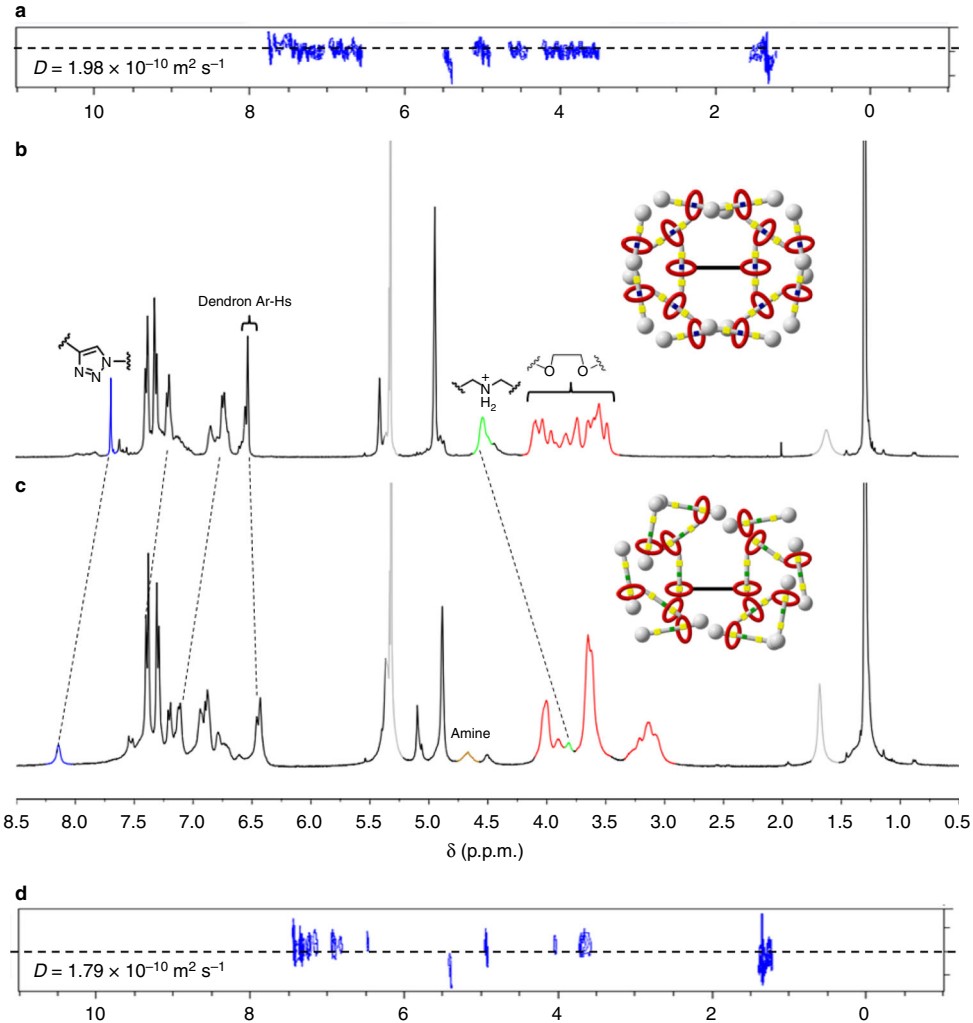

**Fig. 4** NMR spectra of G3 T3B-RDs. DOSY NMR **a**, **d** of G3 and neutral G3 T3B-RDs confirmed the purity of the compounds with the diffusion constant for calculating the hydrodynamic radius. The upfield shift of DBA protons and the downfield shift of triazole protons **b**, **c** indicated the molecular shuttling of the macrocycles from the original DBA to triazole, after the complete deprotonation. $CD_2Cl_2$ was used as the solvent for the NMR analysis

confirming a complete removal of all $PF_6^-$. Since the deprotonated neutral T3B-RDs did not carry any charge, the mass data were obtained from HR-MALDI-TOF. The peak of neutral G1 T3B-RD was observed clearly, but in G2 ($m/z = 9305$), additional masses of fragments were found due to a stronger laser power used. For G3, MALDI-TOF MS spectrum cannot be obtained, possibly because of its high-molecular weight (theoretical $m/z = 20,496$), rendering the ionization step very selective and difficult. DOSY NMR was also used to determine the purity of all deprotonated neutral compounds, and the results showed that only single component was presence, confirming the complete deprotonation with high purity.

**Size information of type III-B RDs**. The hydrodynamic size of the G1–G3 T3B-RDs were characterized by dynamic light scattering (DLS) in $CH_2Cl_2$ (Supplementary Fig. 49). The hydrodynamic size of G1 was determined as $1.80 \pm 0.14$ nm, progressively increasing by generation (G2 = $3.50 \pm 0.21$ nm, G3 = $4.57 \pm 0.81$ nm). The increase of the hydrodynamic size was nearly doubled from one generation to the next. The broadness of DLS peaks are increasing as increasing dendrimer's generation, possibly because of the increasing flexibility in higher generation dendrimers which can adopt more different structures in solution. The diffusion coefficient obtained from DOSY was used to

calculate the hydrodynamic radius and diameter (size) (Supplementary Table 1). For G1 T3B-RD, the determined value was close to DLS, whereas the determined values of G2 and G3 T3B-RDs were a bit smaller than DLS, based on different assumptions. Nonetheless, the increasing trend in radius/size of G1–G3 T3B-RDs was consistent with DLS. The hydrodynamic radii/sizes of deprotonated neutral G1–G3 T3B-RDs were determined by DOSY. As expected, the determined radii/sizes of all deprotonated neutral G1–G3 T3B-RDs showed slightly larger ($\Delta R = 0.09$–$0.14$ nm, ~10%) values than the original ionic state. These results suggested that after the deprotonation of T3B-RDs by base, T3B-RDs could induce a conformational change from the contract state to the extended state. With a higher generation (e.g., G3), the size change increases, because more rotaxanes (14 for G3) inside the macromolecule were switched together, leading to a larger size extension. DFTB+ study indicated that the structure was heavily folded, so as to reach the very compact size measured from experiments (Supplementary Fig. 45).

Atomic force microscopy (AFM) was employed to visualize the morphologies of G1–G3 T3B-RDs (Fig. 8, and Supplementary Figs. 50–55) on mica surface[32,33]. From the tapping mode AFM images, uniform single near-spherical morphologies were observed in all G1–G3 T3B-RDs. Slight aggregation was observed in G1 T3B-RDs. The average height of the G1–G3 T3B-RDs was increased from ~1.15 (G1), ~3.94 (G2), and to ~10.96 nm (G3).

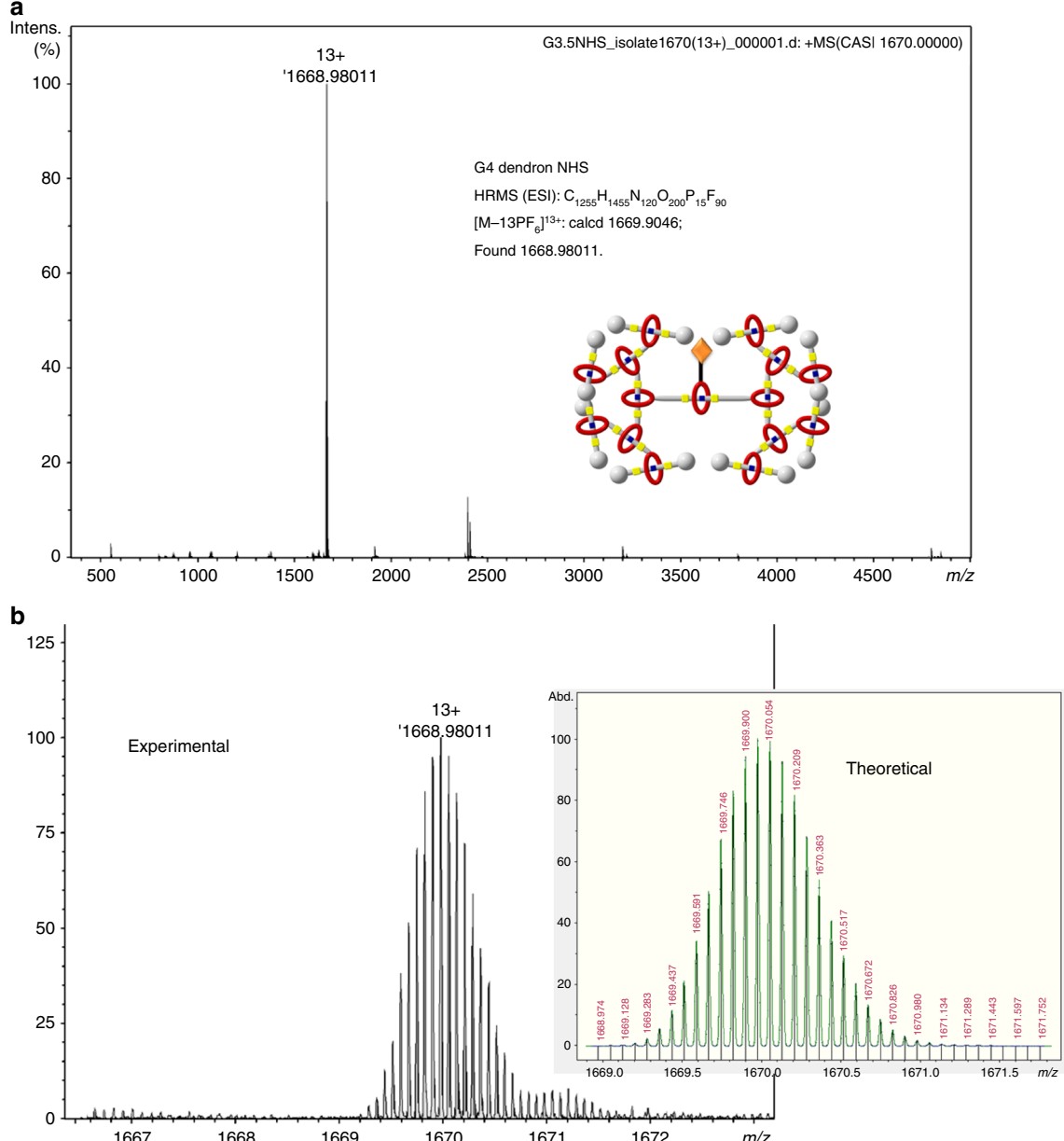

**Fig. 5** High resolution electrospray ionization mass spectrometry (HR-ESI-MS) spectra of G4 dendrons. A single-13$^+$ ion specie was observed in the ESI-MS spectrum **a**, which representing the [M–13PF$_6$]$^{13+}$ ion of G4 dendrons. The isotopic pattern **b** of experimental result was consistent with the theoretical simulation, confirming the successful synthesis of G4 dendrons

The morphologies of neutral G1–G3 T3B-RDs were also be visualized (Supplementary Figs. 53–55). In all neutral G1–G3 T3B-RDs, similar morphologies and the height increase were observed (Supplementary Table 4). The height differences between the ionic and neutral G1–G3 T3B-RDs were ranging from 20 to 22%, height increase progressively from G1 ($\Delta h = 0.23$ nm) to G2 ($\Delta h = 0.87$ nm), and G3 ($\Delta h = 2.29$ nm). Since G3 T3B-RD contains 28 triazoles, the ring shuttling between the triazoles were significant, and that the molecules would tend to form the most stable conformation, avoiding the steric hindrance from dendrons after the deprotonation (relaxed state). As the generation increases, a larger height difference was observed due to the number of switching state was also increased. These T3B-RDs generally possessed some rigidities, whereas they would not be easily flatten on surface.

DOSY, DLS, and AFM were employed to show the size and morphology of T3B-RDs. In G1–G2 T3B-RDs, the three

techniques showed the similar size except G3. AFM measured the rotaxane dendrimer in solid state after a spin coating on mica surface. The spin-coated solid-state dendrimers may not be perfectly spherical molecules. In the case of G3, due to the rigidity with more hydrophobic tertiary butyl aryl groups, the adsorption forces between rotaxane dendrimer and hydrophilic mica surface will decrease with higher generation in the solid state, thus G3 would not be easily flatten on mica surface, giving a higher height than the sizes measured by DOSY and DLS in solution state. By analyzing from these three techniques, a progressive trend of size increase was observed from G1 to G3.

**Responsive switching of type III-B RDs**. We had investigated the acid–base triggered switching of T3B-RDs in solution. The switching processes were monitored by $^1$H-NMR spectroscopy carefully. Triflouroacetic acid (TFA) and 1,8-diazabicyclo(5.4.0) undec-7-ene (DBU) were the acid and base for studies. First, we

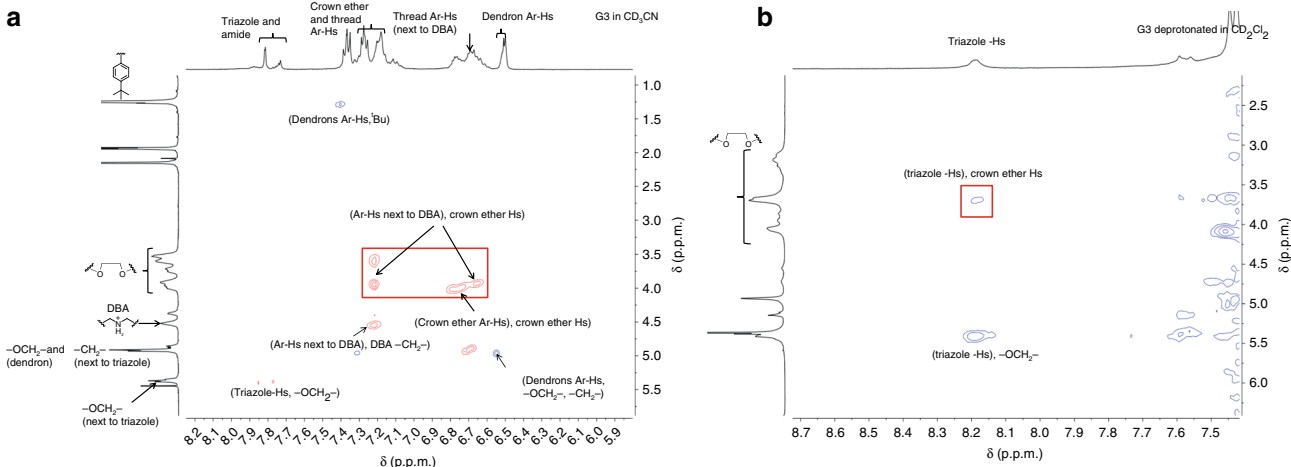

**Fig. 6** NOESY NMR spectra of G3 T3B-RD. The cross peaks of aromatic protons signal **a** of DBA with crown ether glycol protons confirm the position of the macrocycle at DBA, while no cross signal was found between triazoles and crown ether glycol protons. After the deprotonation **b**, a cross peak between triazoles and crown ether protons was observed, indicating the macrocycles encircled the triazoles

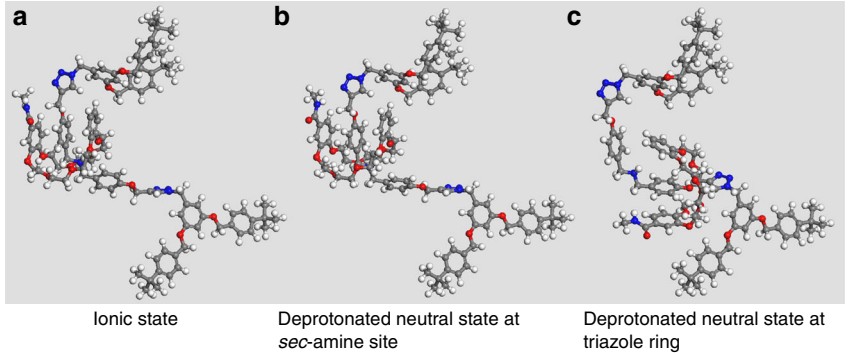

|  |  |  |
|---|---|---|
| Ionic state | Deprotonated neutral state at *sec*-amine site | Deprotonated neutral state at triazole ring |

**Fig. 7** DFTB+ computation model. DFTB+ Optimized truncated structures of [2]rotaxane in three different states **a** ionic state, **b** deprotonated neutral state at the sec-amine site, and **c** deprotonated neutral state at the triazole ring

tested the switching reversibility of all G1–G3 T3B-RDs (Supplementary Figs. 59–61). In the case of G1 T3B-RD (Supplementary Figs. 56 and 59), upon deprotonation by 2.0 equiv of DBU, significant NMR chemical shift was observed. DBA protons $H_l$ were shifted upfield, and the resonance triazole protons $H_h$ were shifted downfield, confirming the shuttling of macrocycle from DBA to the triazoles. Same chemical shifts were observed as described after the BMEP-resin deprotonation. When a slight excess of TFA (2.1 equiv) was added to the deprotonated solution, the whole spectrum was restored back and identical to the original one, indicating the macrocycle shuttled back to the DBA sites. As proved by [1]H NMR, after seven complete cycles, G1 T3B-RD's spectrum can still be restored with only very slight degradation. The DBA protons and triazole protons in G2 and G3 T3B-RDs also experienced the same chemical shifts and restoration as in the case of G1. However, since more DBA sites (6 for G2 and 14 for G3) were presence, much more equivalents of base and acid were needed for triggering the shuttling processes. As a result, large accumulation of salt residue from neutralization somewhat interrupted the switching cycles, with an acceptable degradation after 7 complete cycles.

**Potential application of type III-B RDs**. Our deprotonated neutral T3B-RDs contain a number of sec-amine groups that are free for binding with guest molecules through electrostatic interaction. We hypothesize that those bound molecules can be actively released from the dendrimers after a collective molecular shuttling of macrocycle back to the DBA. As a proof of concept

manner, we use chlorambucil, a small drug molecule with carboxylic acid group to check whether it could be bound with the amine groups within the deprotonated neutral T3B-RDs. [1]H-NMR titration was used (Supplementary Figs. 68–76) to determine the host-guest supramolecular interaction[34–37]. When 0.3 equiv of chlorambucil was titrated to the neutral G1 T3B-RD, the triazole proton $H_h$ was disappeared (peak coalesce). This is because the electrostatic interaction of ammonium and carboxylate blocked the shuttling of the macrocycle between two triazoles, thereby the shuttling of macrocycle between two triazoles became slower in NMR timescale. Computation study was also consistent with the experimental findings (Supplementary Fig. 48). The binding constant ($K_a$) of neutral G1 T3B-RDs with chlorambucil was calculated to be $5.2 \times 10^5 \, M^{-1}$ (Supplementary Fig. 77). For neutral G2 T3B-RD, very similar chemical shifts can be found, and all amino groups in G2 were capable to bind with four guest molecules. The largest G3 T3B-RD can bind with eight guest molecules out of the 14 DBA sites, possibly due to the large steric hindrance inside the core of the molecules by the DB24C8, the guest molecules might only be able to bind on the outer layer of T3B-RDs. Chlorambucil can be unbound from T3B-RDs when an excess of acid was added, as proven by [1]H-NMR spectroscopy, the macrocycles shuttled back to DBA sites and restored the original NMR signals of chlorambucil.

## Discussion
Higher generation (up to G4) type III-B RDs (T3B-RDs) were successfully synthesized and unambiguously characterized by

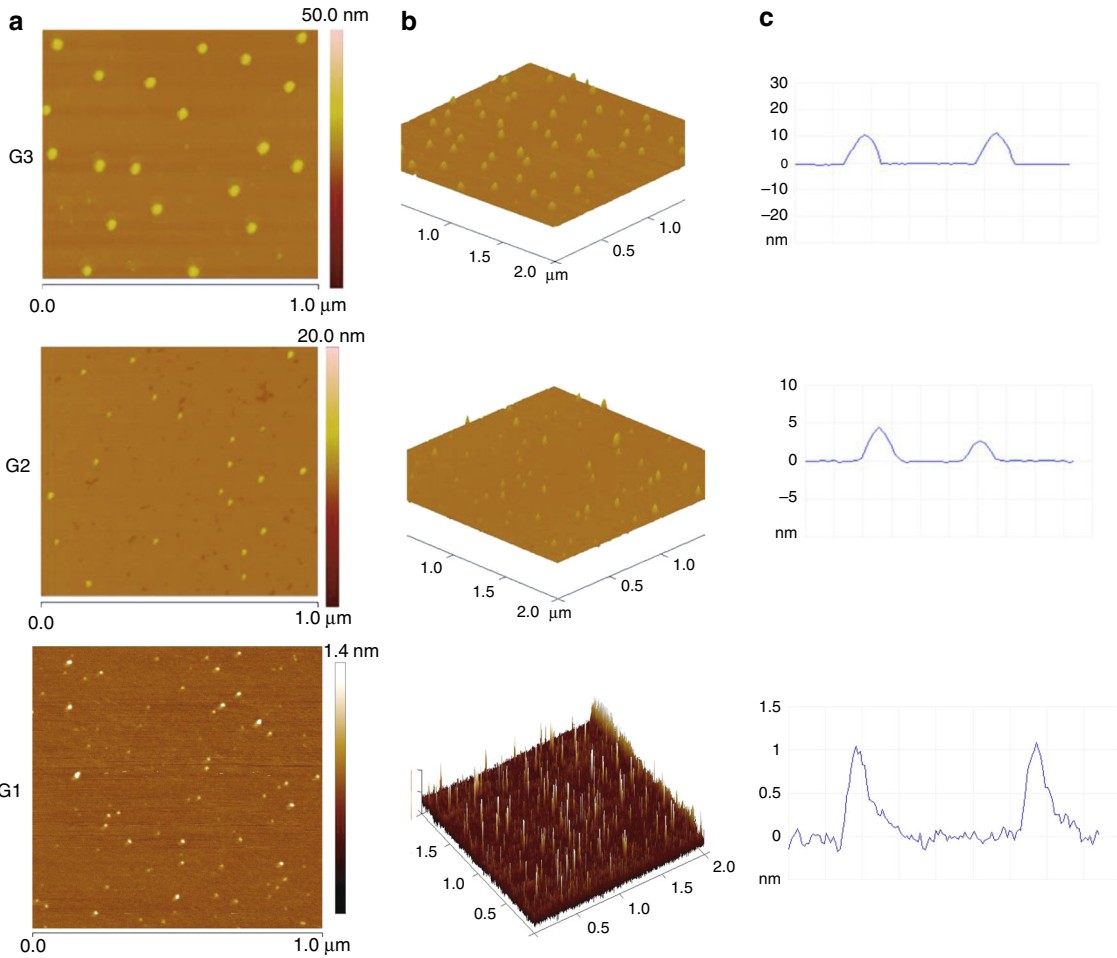

**Fig. 8** AFM images. The AFM images **a**, **b** were performed on mica surface via spin coating, and monodispersed morphologies were clearly observed for all T3B-RDs. The height **c** of the molecules increased gradually from G1 to G3

various spectroscopic techniques, including $^1H$, $^{13}C$, $^{31}P$ NMR, NOESY, and DOSY, as well as HR-MALDI-TOF and HR-ESI mass spectrometries. DLS and AFM were used to determine their structural sizes and mophologies, showing their discrete and rigid nature. A total of six species of dendrimers (G1–G3 and neutral G1–G3) were studied. All T3B-RDs were found to be acid–base switchable, and the triazole can be the site for the molecular shuttling of DB24C8. The sizes of deprotonated neutral G1–G3 T3B-RDs were slightly larger than G1–G3 demonstrated by the DOSY study, showing the observable and characterizable, reversible contract-extend state within the T3B-RDs. Chlorambucil was used as a guest model in the host-guest binding system with the neutral T3B-RDs, and it was shown that the dendrimers can bind a number of guest molecules within their structures. The largest deprotonated neutral G3 T3B-RD can bind with 11 guest molecules. We have developed a facile synthetic approach of T3B-RDs, demonstrated their potential applications in guest binding by molecular shuttling in 3D manner. The synthesis of even higher generation T3B-RDs is currently underway in our laboratory. This study provides important information for chemist to further develop different versions and higher generations of RDs with various properties and potential applications in smart-molecular machines, materials, and nanotechnology.

## Methods

**Preparation of G3 [15] RDs**. G3 [8]Rotaxane dendron-$N_3$ (0.30 g, 0.027 mmol), and G3 [8]rotaxane dendron-acetylene (0.30 g, 0.027 mmol) were dissolved in 5 mL degassed $CH_2Cl_2$. Cu(MeCN)$_4$PF$_6$ (0.03 g, 0.08 mmol) and $^i$Pr$_2$EtN (0.02 mL, 0.12

mmol) were added. The reaction was stirred for 3 days at room temperature. The reaction mixture was diluted with 30 mL $CH_2Cl_2$, and extracted with (1) 20 mL 2 M NaCN$_{(aq)}$, (2) 20 mL 1 M HCl$_{(aq)}$, and (3) NH$_4$PF$_{6(aq)}$. The combined organic layers were dried over anhydrous MgSO$_4$ and concentrated under vacuum to give a yellow solid. The yellow solid was purified by column chromatography (SiO$_2$; $CH_2Cl_2$/MeOH 20:1 → 10:1) yielding a pale yellow solid (0.43 g, 71%).

**Preparation of G4 [16] Rotaxane Dendron-NHS**. Thread H·PF$_6$ (0.014 g, 0.031 mmol), DB24C8-OSu (0.080 g, 0.13 mmol), and G3 [8]rotaxane dendron-N$_3$ (0.70 g, 0.062 mmol) were dissolved in 8 mL degassed $CH_2Cl_2$. The reaction mixture was stirred for 2 h before the addition of Cu(MeCN)$_4$PF$_6$ (0.10 g, 0.26 mmol). The reaction was then stirred for 6 days at room temperature. The reaction mixture was diluted with 70 mL $CH_2Cl_2$, and extracted with (1) 50 mL 2 M NaCN$_{(aq)}$, (2) 50 mL 1 M HCl$_{(aq)}$, and (3) NH$_4$PF$_{6(aq)}$. The combined organic layers were dried over anhydrous MgSO$_4$ and concentrated under vacuum to give a yellow solid. The yellow solid was purified by column chromatography (SiO$_2$; $CH_2Cl_2$/EtOAc 1:1 → Acetone → Acetone with NH$_4$PF$_6$ (0.30 gL$^{-1}$)) yielding a pale yellow solid (0.30 g, 41%).

**Preparation of neutral RDs**. G(n) RDs were dissolved in 1 mL CD$_3$CN, BEMP resin (2-tert-butylimino-2-diethylamino-1,3-dimethylperhydro-1,3,2-diazapho-sphorine, polymer-bound (Sigma-Aldrich)) was added and stirred overnight at room temperature. $CH_2Cl_2$ was added to re-dissolve the deprotonated neutral rotaxane dendrimer, BEMP resin was filtered, and the combined organic layers were concentrated under vacuum to afford the deprotonated neutral G(n) rotaxane dendrimer as a yellow solid in quantitative yield.

**Data availability**. The authors declare that all the data supporting the findings of this study are available within the paper and its Supplementary Information files. All data are available from the authors upon reasonable request.

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

## Acknowledgements

This study was supported by the Area of Excellence Scheme (AoE-P03/08) from the University Grants Committee of Hong Kong. This work was partly supported by the Collaborative Research Fund of Hong Kong Research Grants Council (Project No. C2014-15G) and the Inter-institutional Collaborative Research Scheme, a grant sponsored by the Research Committee of HKBU (RC-ICRS/15-16/01). We also acknowledge the computing resources of the Tianhe2-JK cluster at the Beijing Computational Science Research Center and of the Tianhe2 cluster at the National Supercomputer Center in Guangzhou, China. ICTS is supported by the Institute of Creativity of HKBU, which is sponsored by the Hung Hin Shiu Charitable Foundation.

## Author contributions

C.-S.K. and K.C.-F.L. conceived and designed the experiments. C.-S.K. and K.C.-F.L. completed the synthesis and characterization. R.Z. and M.A.V.H. performed the DFT calculation. Z.C. performed the mass spectral analysis. C.-S.K., M.A.V.H. and K.C.-F.L. co-wrote the manuscript and analysed the data. K.C.-F.L. directed the study.

## Additional information

**Competing interests:** The authors declare no competing financial interests.

