## [Peer Review File · Nature Communications]

Reviewers' comments:

Reviewer #1 (Remarks to the Author):

In this manuscript, Leung and coworkers demonstrated the successful synthesis of Type III-B rotaxanes dendrimers up to third generation and even fourth generation dendron. It should be noted that the construction of Type III series of rotaxane dendrimers is extremely challenging, especially those with stimuli-responsive behavior. As I know that there have been very few successful examples of Type III rotaxane dendrimers with high generation. There is no doubt that the authors made a great advance in this area. Moreover, the authors presented the detailed investigations on the acid-base controlled switching behaviors of the resultant rotaxane dendrimers. According to this work, the controllable 3-D molecular switching of rotaxane dendrimers has proven to be feasible, which opens a new avenue to the novel molecular machine. Definitely, the chemistry presented in this manuscript will receive considerable attention from the broad readership of Nature Communications. I do love this chemistry and strongly recommend it to be accepted after all issues listed below have been addressed.

1. Have the authors tried the synthesis of the functionalized G4 dendrons with azide and acetylene moieties and the targeted G4 rotaxane dendrimer? If so, the authors should provide the primary results about the synthetic results. I understand that it is very difficult to get G4 rotaxane dendrimer. But it will be very helpful for readers to understand better about this chemistry if the authors provide some primary results;
2. In the case of the purity of the ionic and neutral G1–G3 rotaxane dendrimers, only DOSY experiments were performed in the manuscript. However, HPLC or GPC analysis might be very helpful to check the purity;
3. In the Figure 8, the AFM images showed the morphology and size information of the rotaxane dendrimers G1-G3. However, the morphology of the G1 is inhomogeneity, which might be due to the slight aggregation as claimed by the authors. In order to give a better description of the increase of size, a uniform morphology is preferred. Furthermore, TEM measurements should also be performed to visual the morphology and size information;
4. In the supporting information, the peak of G3 in the DLS analysis (Figure S23) was too broad and not a single peak, which was not accurate enough to give any size information;

5. The sizes of the resultant rotaxane dendrimers were measured by three different analysis means, i.e, DOSY, AFM and DLS. However, the values of the same generation strongly varied in different analysis methods. For example, by AFM analysis, the average height of G3 is ~10.96 nm, whereas DLS measurement showed ~4.57 nm, and DOSY displayed ~1.35 nm. A reasonable explanation of the difference should be provided;
6. In order to display a more intuitive descriptions of the size change after deprotonation, AFM and DLS analysis of the corresponding neutral rotaxane dendrimers should be performed;
7. Simply based on the NMR titration, how to confirm the exact numbers of guest molecules binding with the rotaxane dendrimers?
8. As a critical parameter on host-guest chemistry, the binding constants of the rotaxane dendrimers G1-G3 towards the guests are very necessary.

Reviewer #2 (Remarks to the Author):

In the previously published paper (W. Wang, L.-J. Chen, X.-Q. Wang, B. Sun, X. Li, Y. Zhang, J. Shi, Y. Yu, L. Zhang, M. Liu and H.-B. Yang, Proc. Natl. Acad. Sci. U. S. A., 2015, 112, 5597), Yang and coworkers has well established the synthesis, characterization, and functionalization of higher-generation (up to fourth-generation) organometallic rotaxane branched dendrimers using pillar[5]arene and sequential coupling–deprotection–coupling processes. Although this manuscript use the another host-guest interaction and coupling reactions, the point (fourth-generation) is still in absence of innovation. Therefore, this manuscript should not publish in Nature Communications.

Reviewer #3 (Remarks to the Author):

I would like to commend on this paper according to the guideline for reviewers.

1. What are the major claims of the paper?

-The author successfully synthesized and fully characterized a series of type III-B rotaxane dendrimers (dendrons) up to 4th generation. They also observed the pH controlled switching behaviour and their preliminary application for controlled drug release.

2. Are they novel and will they be of interest to others in the community and the wider field?

-Synthesis of mechanically locked rotaxane dendrimers is a very challenging task, especially the type III-B dendrimers. This work is of particular interest to the general readers in supramolecular chemistry. It is also interesting for readers in structural organic chemistry and organic materials science.

3. Is the work convincing?

-I have carefully checked all the characterization data (1D and 2D NMR, HR MS, LS, AFM , etc.), I think all the compounds are well characterized and it is convincing.

4. Do you feel that the paper will influence thinking in the field?

-Yes, the synthetic strategy is new, and the switching behaviour and the preliminary test give some new insight into the potential application of using rotaxane dendrimers or hyperbranched polymers for controlled drug release applications.

5. Appropriateness and validity of any statistical analysis, as well the ability of a researcher to reproduce the work.

-The data are well analysed and is reproducible.

I only have minor issue. The authors talked about the after deprotonation, the crown ether will stay at the triazole unit based on the theoretical calculations. It would be nice if they can further elaborate this, for example, what is the energy difference between the two possible states?

The paper was well written and the supporting information is clear.

Reviewer 1

We greatly thank and appreciate the reviewer acknowledged the novelty of our work, also giving us some valuable feedback and useful suggestions to modify and enhance the quality of the manuscript. We have now taken into account carefully of all suggestions and concerns given by the reviewer, and have modified the manuscript accordingly.

1. Have the authors tried the synthesis of the functionalized G4 dendrons with azide and acetylene moieties and the targeted G4 rotaxane dendrimer? If so, the authors should provide the primary results about the synthetic results. I understand that it is very difficult to get G4 rotaxane dendrimer. But it will be very helpful for readers to understand better about this chemistry if the authors provide some primary results.

Response: We have tried the synthesis of the functionalized G4 dendrons with azide and acetylene moieties and the targeted G4 rotaxane dendrimer. However, the resulted data are too preliminary, wherein we only got the ^1H NMR and the ^{13}C NMR spectroscopic results. The preliminary data are shown below:

G4 [16]rotaxane dendron-azide

G4 [16]rotaxane dendron-acetylene

In order to fully study the G4 [31]rotaxane dendrimer, it requires a much longer or unknown time to finish up all repeated synthetic steps and characterization, including 1-D NMR, 2-D NMR, MS, AFM, DLS, GPC analysis, etc. Therefore, at this stage, we tend to not disclose the preliminary results on the G4 [31]rotaxane dendrimer. In page 14 of the revised manuscript, we have added a sentence, stating that we are continuing the further investigation on G4 and higher generation rotaxane dendrimers.

2. In the case of the purity of the ionic and neutral G1–G3 rotaxane dendrimers, only DOSY experiments were performed in the manuscript. However, HPLC or GPC analysis might be very helpful to check the purity.

Response: We agreed with the reviewer concerning about the purity of the rotaxane dendrimers with only DOSY experiments. We then performed an additional GPC analysis for the neutral G1 rotaxane dendrimer. From the GPC chromatogram, we can observe a single peak, further confirming the purity of the compounds.

For other higher generation neutral rotaxane dendrimers, unexpectedly due to the strong adsorption with the GPC columns, we cannot obtain satisfactory results.

For the ionic G1–G3 rotaxane dendrimers, we have not performed the GPC/HPLC analysis. Since the neutral rotaxane dendrimers have already strongly adhered on the columns and that the ionic rotaxane dendrimers carry high charges, they may not be able to elute out from the column and would cause column damage. Therefore, for the cases of ionic G1–G3 rotaxane dendrimers, we think that the combinations of clear 1-D NMR (^1H , ^{13}C , ^{31}P), 2-D NMR (NOESY and DOSY) spectroscopy and ESI-MS analysis can confirm the purity of the targeted compounds.

3. In the Figure 8, the AFM images showed the morphology and size information of the rotaxane dendrimers G1-G3. However, the morphology of the G1 is inhomogeneity, which might be due to the slight aggregation as claimed by the authors. In order to give a better description of the increase of size, a uniform morphology is preferred. Furthermore, TEM measurements should also be performed to visual the morphology and size information.

Response: We agreed with the reviewer about the inhomogeneity and aggregation of AMF image (Figure 8) of G1 rotaxane dendrimer. We then carefully prepared and retried the G1 AFM sample analysis. We clearly observed the monodispersed uniform G1 rotaxane dendrimer on mica surface, and the Figure 8 and its discussion (page 11) were modified accordingly.

We have also tried the TEM analysis for rotaxane dendrimers, however, we cannot observe the useful morphology and size information, possibly due to (1) relatively low resolution to observe objects of 1-10 nm and (2) the fragile organic composition of rotaxane dendrimers which cannot be readily visualized using the high voltage electron beam technique (*Macromolecules* **1998**, *31*, 6259–6265). The TEM results are shown below:

Since AFM has a better resolution towards the pure organic small molecules of 1-10 nm, we therefore believed that the images obtained from AFM are relatively reliable to obtain the morphology and size information of all ionic and neutral rotaxane dendrimers.

4. In the supporting information, the peak of G3 in the DLS analysis (Figure S23) was too broad and not a single peak, which was not accurate enough to give any size information.

Response: G3 is the largest III-B rotaxane dendrimer and flexible with the hyperbranched structures and Fréchet-type aryether dendrons. When G3 was

dissolved in solution (DCM) state, G3 will keep changing its shape and give the DLS analysis a bit broad signal, in comparison to G1 and G2. Similar broadening effect in DLS analysis could be observed from other reported higher generation dendrimer molecules (*Bioconjugate Chem.* **2004**, *15*, 1221–1229; *Macromolecules* **2014**, *47*, 2199–2213; *Proc. Natl. Acad. Sci.* **2015**, *112*, 5597–5601). This point has been discussed in the main text (page 10) “The broadness of DLS peaks are increasing as increasing dendrimer’s generation, possibly because of the increasing flexibility in higher generation dendrimers which can adopt more different structures in solution”.

5. The sizes of the resultant rotaxane dendrimers were measured by three different analysis means, i.e, DOSY, AFM and DLS. However, the values of the same generation strongly varied in different analysis methods. For example, by AFM analysis, the average height of G3 is ~10.96 nm, whereas DLS measurement showed ~4.57 nm, and DOSY displayed ~1.35 nm. A reasonable explanation of the difference should be provided.

Response: We thank to the comment from the reviewer about the unclear description on the sizes of rotaxane dendrimer using three different techniques. For G1–G2, the three techniques showed the similar size except G3. In the case of G3, the size from DLS is about ~4.57 nm, DOSY is about ~2.7 nm, while AFM showed ~10.96 nm in height. The DOSY result of the high molecular weight component will give a broad NMR signal thus having a relatively smaller size. Whereas, the DLS would give better data for high molecular weight molecules because of the scattered light intensity is proportional to the sixth power of the molecular diameter. Both techniques measure the average diffusion coefficients of equilibrated particles in solution state.

While in AFM, it is measured the rotaxane dendrimer in solid state, after the spin coating on the mica surface. In the case of G3, due to the rigidity bring by the mechanical bonds, as well as the high generation, the adsorption forces between rotaxane dendrimer and mica surface will decrease with higher generation, thus G3 would not be flatten (nor spherical) on mica surface, giving a higher height than DOSY and DLS. (*Langmuir* **2000**, *16*, 5613–5616.)

From these three techniques, we can observe a general increase trend of size increase from G1–G3 gradually. This point has been discussed in pages 11-12.

6. In order to display a more intuitive descriptions of the size change after deprotonation, AFM and DLS analysis of the corresponding neutral rotaxane dendrimers should be performed.

Response: We agreed with the reviewer, concerning the characterization of size change of rotaxane dendrimers after the deprotonation using AFM and DLS. We study the AFM of neutral G1–G3 rotaxane dendrimers, the results showed that the heights in neutral G2 and G3 rotaxane dendrimers were higher than the ionic ones, while neutral G1 showed a similar height. The below paragraph and new figures were added to the revised manuscript (pages 11-12) and SI (S27-29). “The morphologies of *neutral* G1–G3 **T3B-RDs** were also be visualized. (Supplementary Fig. S27–S29) In *neutral* G1, similar height (1.23 ± 0.02) was observed, as there were only four shuttling triazole stations for the ring after deprotonation and having a relatively small molecular weight. The height difference between the ionic and neutral G1 rotaxane dendrimer is 7%. In contrast, the heights of *neutral* G2 ($\Delta h = 0.87$ nm, 22%) and *neutral* G3 ($\Delta h = 5.89$ nm, 54%) showed higher heights than the original ionic states. Since G2 rotaxane dendrimer contains 12 triazoles while G3 contains 28 triazoles, the ring shuttling between the triazoles are significant, and that the molecules would tend to form the most stable conformation, avoiding the steric hindrance from dendrons after the deprotonation (relaxed state). As the generation increases, a larger height difference was observed due to the number of switching state was also increased. These **T3B-RDs** generally possessed some rigidity, whereas they would not be easily flatten on surface.”

For DLS analysis, we have tried several conditions in the analysis of neutral G1–G3 rotaxane dendrimers, however, no satisfactory results were obtained. Since the (n)triazole in the rotaxane dendrimers were equivalent to each other, the macrocycle will keep shuttling between (n)triazole, together with flexibility of the molecules itself in solution state. As DLS measurement depends on the scattered light radiation intensity decay as a function of time, when the particles dispersed in solution, the rotaxane dendrimer (particle) size will keep changing, thus resulting the instability in DLS analysis. More intuitive descriptions of the size change after deprotonation are now realized by the new AFM analysis in addition with the DOSY NMR data.

7. Simply based on the NMR titration, how to confirm the exact numbers of guest molecules binding with the rotaxane dendrimers?

Response: We thank the reviewer to pointing out the unclear explanation of molecules binding with the rotaxane dendrimers based on NMR titrations. In terms of determination of the number of guest molecules binding with rotaxane dendrimers, we compared their NMR chemical shifts of the original (ionic) dendrimer and the guest-bound dendrimers.

In G1 dendrimer's NMR titration (Figure S42), when titrating up to 2.0 equivalents of guest molecules, the unique hydrogen-bonded **DBA** proton signals are still not restored (green color), meaning that the macrocycle is still located at the triazole, while the **DBA** interacts with the carboxylate. Once excess acid (guest) was added, the carboxylate reprotonated to carboxylic acid and the macrocycle moved from triazoles back to the original site. Therefore, G1 could bind with two guest molecules by observing from NMR titration. In the case of G2 dendrimer's NMR titration (Figure S44), we could observe the unique DB24C8-DBA binding signals after the addition of 4.0 equivalents of guest molecules, therefore one G2 dendrimer was able to bind with 4 guest molecules. In G3 dendrimer's NMR titration (Figure S46), similar peak shifts were used for the determination of guest molecules binding to the rotaxane dendrimers, and 8 guest molecules were bound. In conclusion, guest substrates can be bound to the amine sites at the outer layer of the dendrimers. This point has been discussed in the revised manuscript (page 13 and figures S43-S47).

8. As a critical parameter on host-guest chemistry, the binding constants of the rotaxane dendrimers G1-G3 towards the guests are very necessary.

Response: We agreed with the point raised by the reviewer about the importance of binding constant. In the case of neutral G1, it binds with two chlorambucil molecules through electrostatic interaction between the ammonium and carboxylate. We then use the chemical shift of the triazole proton for calculating the binding constant with 1:2 host-guest fitting model by the online software BindFit v0.5 demonstrated by Throdarson *et al.* This point has been discussed in the revised manuscript (page 13 and figure S48).

The experimental chemical shift was in agreement with the calculated 1:2 binding

model, and the binding constant (K_a) was calculated to be $5.2 \times 10^5 \text{ M}^{-1}$ by BindFit v0.5. In the case of G2 and G3, as it is a multiple binding with the guest molecules, the binding constants between the guest molecules on the surface, or inside the core will be different, thus it is complicated to calculate the binding constants of G2 and G3. We believed that the binding constant from G1 could give a brief insight about the binding strength of rotaxane dendrimers with guest molecules.

Reviewer 2

In the previously published paper (W. Wang, L.-J. Chen, X.-Q. Wang, B. Sun, X. Li, Y. Zhang, J. Shi, Y. Yu, L. Zhang, M. Liu and H.-B. Yang, Proc. Natl. Acad. Sci. U. S. A., 2015, 112, 5597), Yang and coworkers has well established the synthesis, characterization, and functionalization of higher-generation (up to fourth-generation) organometallic rotaxane branched dendrimers using pillar[5]arene and sequential coupling–deprotection–coupling processes. Although this manuscript use the another host-guest interaction and coupling reactions, the point (fourth-generation) is still in absence of innovation. Therefore, this manuscript should not publish in Nature Communications.

Response: We thank the comment by the reviewer concerning the innovation compared with the published result of type III-A organometallic rotaxane dendrimers by Yang *et al* in 2015. In our work, we have first successfully synthesized up to third generation (G3) *type III-B* rotaxane dendrimers and a fourth generation *type III-B* dendron. By the definition from Kim *et al.*, *Top. Curr. Chem.* **228**, 11–140 (2003) and Stoddart *et al.*, *The Nature of the Mechanical Bond: From Molecules to Machines* (Wiley, New Jersey, 2017), type III-A rotaxane dendrimer is a dendritic polyrotaxane incorporating mechanical bonds in between the branching points on a dendrimer scaffold, while type III-B rotaxane dendrimer is having mechanical bonds constituted the branching points. They are two completely different architectures.

The general structure of the two rotaxane dendrimers:

Type III-A rotaxane dendrimer

Type III-B rotaxane dendrimer

The work presented by Yang *et al* on type III-A rotaxane dendrimers is different from our type III-B rotaxane dendrimers, in terms of structures, nature (organometallic and organic) and synthetic approaches (divergent and convergent). Also, in our work on higher generation type III-B rotaxane dendrimers, we have demonstrated the pH responsive switching process and the size difference from the discrete ionic and neutral G1–G3 rotaxane dendrimers, where the dendrimers synthesized by Yang *et al* cannot be achieved. We also summarized the differences of our work in comparison to Yang’s work in the following table.

	PNAS 2015, 112, 5597.	Our Work
Structure	Type III-A	Type III-B
Nature	Organometallic	Organic
Synthetic Approach	Divergent	Convergent
Protecting Group	Trimethylsilyl (TMS)	Free
Stimuli Responsive	–	pH responsive
Size Change	–	Yes
Substrate Binding	–	Chlorambucil
Surface modification	Ferrocene	–

Therefore, our work on higher generation type III-B rotaxane dendrimers is different from Yang’s published type III-A rotaxane dendrimers in various aspects.

Reviewer 3

I only have minor issue. The authors talked about the after deprotonation, the crown ether will stay at the triazole unit based on the theoretical calculations. It would be nice if they can further elaborate this, for example, what is the energy difference between the two possible states?

Response: We greatly thank and appreciate the reviewer giving us positive comment of our work on type III-B rotaxane dendrimers and the field of supramolecular chemistry. We agreed with the reviewer that we have not clearly showed the energy difference based on calculation. Therefore, concerning about the theoretical calculations on the binding of the crown ether toward triazole unit, we further visualized the theoretical calculations result into an energy diagram, which shown below.

From the energy diagram, we clearly observed the energy state of DB24C8 at triazole was lower than at *sec*-amine, implying the DB24C8 tends to move to the triazole rather than staying at the *sec*-amine after the deprotonation of ammonium demonstrated by the truncated model in DBTB+ calculation. We added this figure in S54.

REVIEWERS' COMMENTS:

Reviewer #1 (Remarks to the Author):

This is a revised manuscript submitted by Prof Leung and co-workers. I have carefully checked the revised version as well as the point-to-point response made by authors. There is no doubt that the authors have made a large step compared to the original manuscript. Just as I commented in the first round evaluation, the chemistry presented in this paper is a very important advance in the field of rotaxane dendrimer, which will receive considerable attention from the broad readership of Nature Communications. So I again strongly recommend it to be published in Nature Communications. My only concern is about the AFM images in the revised version. In all AFM images of neutral rotaxane dendrimers, the particles with different sizes were observed. I fully understand about the AFM results since the possible aggregation of rotaxane dendrimers might occur when the authors prepared the samples for AFM measurements. I suggest that using highly diluted solution for AFM measurement may avoid the possible aggregation of rotaxane dendrimers.

Reviewer #3 (Remarks to the Author):

I am satisfied at the response and revision. The paper now can be published in Nature Communications.

Reviewer 1

This is a revised manuscript submitted by Prof Leung and co-workers. I have carefully checked the revised version as well as the point-to-point response made by authors. There is no doubt that the authors have made a large step compared to the original manuscript. Just as I commented in the first round evaluation, the chemistry presented in this paper is a very important advance in the field of rotaxane dendrimer, which will receive considerable attention from the broad readership of Nature Communications. So I again strongly recommend it to be published in Nature Communications. My only concern is about the AFM images in the revised version. In all AFM images of neutral rotaxane dendrimers, the particles with different sizes were observed. I fully understand about the AFM results since the possible aggregation of rotaxane dendrimers might occur when the authors prepared the samples for AFM measurements. I suggest that using highly diluted solution for AFM measurement may avoid the possible aggregation of rotaxane dendrimers.

Response: Once again, we thank the reviewer 1 for giving us useful suggestions to modify the manuscript and appreciated the importance of our work. We also agree with the reviewer concerning about the new AFM images. In our first revised manuscript, the deprotonated neutral G1 rotaxane dendrimer still have some slight aggregation found in the AFM image. We then tried using a more diluted solution for AFM measurement as suggested by the reviewer 1, and finally, we observed the monodispersed neutral G1 rotaxane dendrimer in new AFM image. We also conducted the G3 rotaxane dendrimer AFM analysis again, and a more uniform morphology was observed in the new images using a highly diluted solution. The new AFM image and the new height information have been updated in Supplementary Figures 53 and 55, and Supplementary Table 4. The descriptions about the neutral AFM images were also updated in the main text.

Reviewer 3

I am satisfied at the response and revision. The paper now can be published in Nature Communications.

Response: We thank the reviewer 3 again for reviewing our manuscript, and supporting the publication of our manuscript in Nature Communications.